# The Impact of the Adjunct Heat-Treated Starter Culture and *Lb. helveticus* LH-B01 on the Proteolysis and ACE Inhibitory Activity in Dutch-Type Cheese Model during Ripening

**DOI:** 10.3390/ani11092699

**Published:** 2021-09-15

**Authors:** Monika Garbowska, Anna Berthold-Pluta, Lidia Stasiak-Różańska, Antoni Pluta

**Affiliations:** Department of Food Technology and Assessment, Division of Milk Technology, Institute of Food Sciences, Warsaw University of Life Sciences-SGGW, Nowoursynowska 159c Street, 02-776 Warsaw, Poland; anna_berthold@sggw.edu.pl (A.B.-P.); lidia_stasiak_rozanska@sggw.edu.pl (L.S.-R.); antoni_pluta@sggw.edu.pl (A.P.)

**Keywords:** cheese model, heat-treated starter, proteolysis, ripening, *Lb. helveticus* LH-B01, ACE inhibitory activity

## Abstract

**Simple Summary:**

Adjunct cultures are used in cheesemaking to improve flavour characteristics and accelerating cheese ripening. Different adjunct cultures are capable of producing enzymes with the specificity to hydrolyze caseins, leading to the release of various bioactive compounds. We studied the effect of adjunct heat-treated starter XT–312 and a cheese culture *Lb. helveticus* LH-B01 on selected physicochemical, microbiological properties, and on proteolysis in cheese models. Additionally, the effect of adjunct cultures on ACE inhibitory activity during ripening was determined. The application of adjunct cultures may be used as functional ingredients in Dutch-type cheese to maintain sufficient bioactive properties and improve proteolysis.

**Abstract:**

Adjunct cultures are used in cheesemaking to improve the sensory characteristics of the ripened cheeses. In addition, it is known that different adjunct cultures are capable of producing enzymes with the specificity to hydrolyze caseins, leading to the release of various bioactive compounds (bioactive peptides, amino acids, etc.). The objective of this study was to evaluate the effect of adjunct heat-treated starter XT–312 and a cheese culture *Lb. helveticus* LH-B01 on the proteolytic activity and angiotensin converting enzymes inhibitors (ACE) in cheese models during ripening. Seven different cheese models were evaluated for: proteolytic activity using the spectrophotometric method with ortho-phthaldialdehyde (OPA), soluble nitrogen (SN), trichloroacetic acid-soluble nitrogen (TCA-SN) phosphotungstic acid-soluble nitrogen (PTA-SN), total nitrogen (TN), pH, contents of water, fat, as well as for total bacteria count (TBC), count of *Lactococcus* genus bacteria, count of *Lb. helveticus*, and number of non-starter lactic acid bacteria (NSLAB). Presence of adjunct bacterial cultures both in the form of a cheese culture LH-B01 and heat-treated XT–312 starter promoted primary and secondary proteolysis, which resulted in acceleration of the ripening process. ACE inhibitory activity and proteolytic activity was the highest throughout of ripening for cheese model with LH-B01 culture. The cheese models with the adjunct heat-treated starter were characterized by lower TBC, NSLAB and lower count of *Lactococcus* genus bacteria during ripening, compared to control cheeses.

## 1. Introduction

Proteolysis is one of the most crucial biochemical processes that play the main role in the development of typical traits of ripening cheeses. The proteolytic system of lactic acid bacteria (LAB) has a significant effect on their growth and affects the formation of flavor compounds in fermented milk drinks. Both the starter lactic acid bacteria (SLAB) and non-starter lactic acid bacteria (NSLAB) are significant sources of proteinases and peptidases that influence proteolysis during cheese ripening, which results in the synthesis of polypeptides, peptides, free amino acids, and other related compounds. The proteolytic system of cheese microbiota plays a key role in the development of its basic organoleptic traits [1]. Most investigations addressing the proteolytic system of LAB have concerned bacteria of the genus *Lactococcus*, whereas the proteolytic system of lactic acid bacilli was less intensively studied. Among the LAB, *L. lactis* is the most often applied component of starters used in cheese making. A few selected strains of *L. lactis* are usually applied as the main components of the basic starters used in cheese manufacture, owing to their desirable technological properties. A secondary starter or an adjunct culture are often added in the production process of cheeses in order to accelerate their ripening, to impart them specific properties, or to intensify their taste profile. In turn, the adjunct cultures may contribute to the development of untypical or extrinsic tastes and aromas (off-flavor) that are classified as cheese defects. This points to the importance of throughout analysis and selection of potential adjunct cultures before their application on the industrial scale [2]. The appropriate fermentation activity of starters determines the apt treatment of milk curds, cheese formation and, to a large extent, its final characteristics. This activity cannot, however, be too high since it would cause excessive acidification of cheese bulk already at the initial stages of production and might also contribute to the development of cheese defects linked with too rapid proteolysis. The weakening of the fermentation activity of adjunct LAB cultures, with their proteolytic activity being preserved, may be most easily achieved through heat treatment or freezing [3,4,5]. The heat treatment of adjunct starter (at the appropriate temperature and elongated time of heating) may allow obtaining desirable proteolytic activity, which is beneficial from the viewpoint of accelerating cheese ripening at preservation or development of its sensory traits [6].

Among the LAB, the use of *Lactobacillus helveticus* is claimed to be the beneficial in cheese proteolysis as well as in the formation of bioactive compounds from casein (especially these with antihypertensive, antioxidant and antimicrobiological activities) and in reducing the risk of colon cancer incidence. Bacteria of this strain are even perceived as immunomodulators in fermented milk and cheese [7,8]. They are also characterized by high diversity of protease genes and by the presence of one to four cell envelope proteinases (CEPs) [9]. The use of different strains of a thermophilic culture *Lb. helveticus*, was initially proposed for the manufacture cheeses with the aim of favouring the development of the flavour of ripened cheeses through hydrolysis of bitter-tasting peptides [10,11,12]. The use of adjunct cultures from the genus *Lactobacillus* in cheese making may have a beneficial effect on the control of the ripening process and increased counts of bacterial flora produced by NSLAB [13].

The characteristics of proteolytic properties of *Lb. helveticus* and other adjunct starters is very important and valuable owing to the possibility of using such cultures as industrial starters that affect the structure, taste, and aroma of cheeses. In addition, the specificity and enzymatic activity of proteinases and peptidases of such adjunct cultures may also influence the release of bioactive peptides with beneficial health-promoting properties, which may be of significance in the production of functional dairy products (milk drinks or cheeses). Bioactive peptides released during proteolysis may have different biological activities such as antioxidant or inhibitory of angiotensin converting enzyme (ACE). In cheeses, this biological activity depends on advanced cheese ripening [14].

Considering the above, the goal of this study was to determine the effect of adjunct heat-treated starter XT–312 with relatively high parameters of heat treatment and a cheese culture *Lb. helveticus* LH–B01 on selected physicochemical, microbiological properties, and on proteolysis in model cheeses. Additionally, the effect of adjunct cultures on ACE inhibitory activity during ripening was determined.

## 2. Materials and Methods

The experimental material included seven model cheeses produced with CHN-19^®^ culture (*L. lactis* ssp. *cremoris*, *Leuc. mesenteroides* ssp. *cremoris*, *L. lactis* ssp. *lactis biovar diacetylactis*) used as the basic starter and with XT–312^®^ culture (*L. lactis* ssp. *cremoris*, *Leuc. mesenteroides* ssp. *cremoris*, *L. lactis* ssp. *lactis biovar diacetylactis*, *L. lactis* ssp. *lactis*, *Leuc. pseudomesenteroides*) and *Lb. helveticus* B01^®^ (LH–B01) (Chr. Hansen, Ltd., Czosnów, Poland) applied as an adjunct starter.

### 2.1. Preparation of Cheese Models

Cheese models were prepared in 500-mL sterile centrifuge bottles (Nalgene, Thermo Scientific, Warsaw, Poland) according to the method described in our previous study [15,16].

Seven variants of cheese models were prepared in the study as shown in Table 1.

In these study all cheese models variants were prepared in triplicate and all analysis were performed two times.

### 2.2. Physicochemical Analyses

All analysis were conducted after five weeks of ripening. Water content was determined by oven drying at 102 °C [17]. Fat content was assayed according to the Gerber method [18], whereas total protein (TN) was determined using the Kjeldahl method [19]. Cheese pH was measured at a temperature of 20 °C (Portamess^®^ 900 pH-meter, KNICK).

### 2.3. Proteolysis

The cheeses were determined for the content of soluble nitrogen (SN), trichloroacetic acid-soluble nitrogen (TCA-SN) and 5% phosphotungstic acid-soluble nitrogen (PTA-SN) using the Kjeldahl method [20] and expressed as a percentage of TN. All determinations were made after one, three, and five weeks of ripening.

Cheese proteolysis was determined using the method with o-phthaldialdehyde (OPA), which is based on the reaction of α-amine groups with ortho-phthaldialdehyde (Sigma-Aldrich, Poznań, Poland) and β-mercaptoethanol (Sigma-Aldrich, Poland) at pH 9.0 in the presence of sodium dodecyl sulfate (SDS) (Sigma-Aldrich) after one day, and two, three and five weeks of ripening [21] The proteolytic activity of all cheese models was determined using 150 μL of the TCA filtrate with 3 mL of o phthaldialdehyde reagent (OPA) according to the method of Church et al. [21]. Absorbance was measured after mixing (2 min incubation, room temperature, 340 nm) using Genesis 10S UV-VIS Spectrophotometer (Thermo Fisher Scientific, Warsaw, Poland).

### 2.4. Determination of the ACE-Inhibitory Activity of Dutch-Type Cheese Models

Water-soluble cheese extracts were obtained according to Garbowska et al. [16] with slight modifications. Five grams of cheese were homogenized (Stomacher Lab-Blender 80, Gemini B.V., Apeldoorn, The Netherlands) for 2 min, after adding 10 mL of deionized water. The pH of the mixture was adjusted to 4.6 with 1.0 M HCl (Sigma-Aldrich), maintained at room temperature for 30 min. Centrifugation (MPW-352R centrifuge, MPW Ltd., Warsaw, Poland) was done at 4500× *g* for 30 min at 4 °C to separate the soluble portion. Supernatant was removed and filtered by Whatman filter No.1 (GE, Medical Systems Polska Ltd., Warsaw, Poland). The supernatant was centrifuged again at 4500× *g* for 15 min, and the clear supernatant was filtered by glass microfiber filter paper (1.0 µm pore size, GE). The filtrate was stored at –40 °C until analysis.

ACE inhibitory activity of each ripening point of cheese model were tested using angiotensin-converting enzyme from rabbit lung, (0.1 U mL^−1^) following the method described in our previous study Garbowska et al. [15,16]. All reagents came from Sigma-Aldrich. Results were calculated according with the:$$\mathrm{IACE}\;\left[\%\right]=\left(1-\frac{C-D}{A-B}\right)\times 100\%$$
where A is the absorbance with ACE and without the sample, B is the absorbance without ACE and with the sample, C is the absorbance with ACE and with the sample, and D is the absorbance with the sample but without ACE.

### 2.5. Microbiological Analysis

For the analysis of total bacteria count (TBC), *Lactococcus* spp., *Lb. helveticus* and non-starter lactic acid bacteria (NSLAB) counts, a 10 g test portion was taken under sterile conditions from each cheese model for microbiological analysis. The sample was transferred into a sterile Bag Page with a full-surface filter (Interscience, Argenta Ltd., Poznań, Poland), poured with 90 mL of a sterile 2% sodium citrate solution (Avantor Performance Materials, Gliwice, Poland) having the temperature of 37 °C, and homogenized (Stomacher Lab-Blender 80). Thus prepared first dilutions of cheese samples were used to prepare serial dilutions in sterile Peptobak pepton (BTL, Łódź, Poland) solution. All analyses were carried out in the fourth week of cheese ripening. Determination of TBC was conducted on Plate Count Skim Milk Agar PCSMA culture medium (Merck, Poland), whereas counts of *Lactococcus* spp., *Lb. helveticus* and NSLAB were determined on M-17 Agar (Merck, Poland), MRS Agar (Merck, Poland) and Rogosa Agar (Merck, Poland), respectively.

### 2.6. Statistical Analysis

Microbiological counts were converted to log CFU/g. Results obtained were analysed using one-way analysis of variance (ANOVA) (Statistica, version 13 software, TIBCO Software Inc., Kraków, Poland). Statistical significance between mean values was determined at *p* < 0.05 using Tuckey’ test.

## 3. Results and Discussion

### 3.1. Physicochemical Analysis

Table 2 summarizes the effect of additional heat-treated XT–312 starter and a cheese culture LH-B01 on the chemical composition of cheese models. No differences (*p* < 0.05) were observed in the main parameters of moisture, protein, fat or in moisture on fat free basis (MFFB) and fat in dry matter (FDM) in cheese models, compared to the control cheeses. The percentage content of fat and water in cheese models fitted within ranges of 17.03–17.50% and 40.24–41.26%, respectively (Table 2). The prepared cheese models were similar in terms of composition to reduced-fat Dutch-type cheese.

The content of total nitrogen (expressed as protein) in cheese models ranged from 30.96 to 31.18% (Table 2). The content of total protein depends mainly on contents of fat and water, and reaches ca. 26% in semi-hard cheeses. Reduced-fat cheeses are, however, characterized by a considerably higher protein content and harder structure compared to full-fat cheeses [22,23].

### 3.2. Proteolysis

Proteolytic changes during cheese models ripening are characterized by primary proteolysis indicated by SN, intermediate proteolysis by TCA-SN and advanced by PTA-SN [24]. Changes in the content of different nitrogen substances over the ripening period are a measure of the extent of cheese ripening [25]. The content of SN (expressed per total nitrogen) in cheese models ranged from 8.69 to 30.56% (Table 3). As shown in Table 3, the content of SN increased progressively over the ripening period in all cheese models.

After one week of ripening of the cheese model with the adjunct starter XT–312 heated to 60, 65, 70 and 75 °C, the content of SN was similar but significantly higher than in the other cheese models. In the case of the cheese models with *Lb. helveticus* LH–B01 addition, the content of soluble nitrogen substances was observed to increase over the ripening period by almost 21.8 percentage points between the first and fifth week of ripening. In the same period (one to five weeks of ripening), in the cheese models produced with the addition of the heat-treated adjunct starter, the greatest increase (by ca. 16.8 percentage points) in the content of SN was determined in cheese models with XT–312 starter heated at 55 °C. The higher heating temperature of the additional XT–312 starter, the smaller SN increase was found between weeks one and five of ripening of the tested cheese models. After five weeks of ripening, a higher content of SN was found in cheese model with *Lb. helveticus*. The *Lb. helveticus* strains may be applied in the production of fermented milk drinks as basic or adjunct starter cultures owing to their capability for milk acidification and synthesis of bioactive peptides and aromatic compounds [26,27]. Because fresh milk does not contain sufficient quantity of free amino acids and low-molecular peptides that are indispensable for the growth of *Lb. helveticus,* these bacteria need an active proteolytic system to initiate the hydrolysis of milk proteins, peptides and amino acids. Next to *L. lactis, Lb. helveticus* is acknowledged as one of the most proteolytic LAB species, which can be explained by the presence of a few genes constituting its proteolytic system. This relatively strong proteolytic system contributes to the release of short peptides and amino acids from the casein matrix and is composed of cell envelope proteinases (CEP) that hydrolyze casein into shorter fragments, transport systems, that enable the uptake of oligopeptides and various intracellular peptidases with diverse, sometimes partially matching, specificity against the pool of free amino acids [26,28]. This may explain more intensive proteolytic transformations occurring during the ripening of model cheeses made with *Lb. helveticus* LH-B01 culture compared to the cheese models from the remaining variants.

The TCA-SN reflects secondary hydrolytic breakdown of casein proteins into simpler, soluble substances [29]. TCA-SN contents of the cheese models obtained in this study continued to increase during ripening (Table 3). Lower values were observed for control and with addition XT–312 heated at 55 °C toward the end of the ripening. After five weeks, higher levels of TCA-SN were observed in cheese model with addition of LH-B01 than those in control cheese model and with heat-treated XT–312 starter. This may be due to the greater extent of primary proteolysis, the products of which may have served as substrates for subsequent hydrolysis by *Lb. helveticus* peptidases. Results from our study agree with previous research, which indicated that after formed by rennet of soluble peptides, which were hydrolysis at a fast rate by bacterial peptidases [30,31,32].

The presence of PTA-SN represents amino acids and very small peptides in the medium, and its level correlates with flavour best in mature cheeses [33]. The PTA-SN values of all cheese models increase from beginning to the end of the ripening. LH-B01 culture-added cheese showed more PTA-SN after three and five weeks of ripening than other cheese models (Table 3); the content of PTA-SN in the remaining cheese models was 2–2.5 times lower. In all cheese models with additional heat-treated XT–312 starter, a similar PTA-SN content was found regardless of the temperature of heat treatment, as well as in the control cheese model. This means that the use of additional heating XT–312 starter does not increase the PTA-SN content in cheese models.

The proteolytic activity was determined in the analyzed cheese models with the OPA method which allows detecting the released α-amino groups. The overall proteolysis increased with the age of ripening in all examined cheeses (Figure 1). The cheese models containing the *Lb. helveticus* culture and variants of cheese models with the adjunct XT–312 starter heated at 70 and 75 °C were characterized by the highest proteolytic activity after five weeks of ripening. In turn, the lowest proteolytic activity was determined in the last week of ripening in control cheeses and in model cheeses with XT–312 adjunct starter heated at 55 °C.

One day after cheese making, the lowest proteolytic activity was shown for the control models of cheeses, whereas the highest one was for the model cheeses with XT–312 adjunct starter heated to 70 °C. In this case, the activity was higher though similar to that determined in the cheeses with the addition of *Lb. helveticus* LH-B01.

It may be noticed that the greatest differences in the proteolytic activity occurred in the analyzed cheese models with adjunct heat-treated XT–312 starter after five weeks of ripening. These differences were increasing along with the increasing temperature of adjunct starter heating. It points to the preserved proteolytic activity of the adjunct starter even after so severe thermal treatment or to the effect of such a starter on bacteria that constitute the basic starter, thus indirectly affecting their proteolytic activity. In studies addressing these issues, the ripening of cheese was accelerated by adjunct starter heating at temperatures ranging from 50 to 72 °C, usually for 10–20 s [4]. Results obtained in the present study indicate that heating at even higher temperature for 15 min effectively accelerated the proteolysis in the analyzed cheese models.

Poveda et al. [1] concluded that Manchego cheeses manufactured with selected autochthonous starter cultures exhibited higher levels of proteolysis than the commercial starter culture cheeses. They found that the cheeses manufactured with the addition of *Lb. paracasei* subsp. *paracasei* (new species name: *Lacticaseibacillus paracasei* subsp. *paracasei*) were characterized by a higher content of free amino acids compared to the other cheeses. The enhanced proteolytic activity in these cheeses was due to nothing but the use of a probiotic culture *Lb. paracasei* subsp. *paracasei* for their manufacture. A similar dependency was observed in our study, because after five weeks of ripening the highest proteolytic activity was determined for the cheese models with the addition of *Lb. helveticus* LH-B01. Bergamini et al. [34] reported that bacteria of the genus *Lactobacillus* had strong proteolytic properties and were applied in cheese making to accelerate the ripening process and to develop aroma compounds. In turn, Garabal et al. [35] achieved a similar proteolytic activity of *L. lactis* and mesophilic relatively heterofermentative lactobacilli.

Farkye et al. [36] determined the highest mean concentration of free amino acids in cheese models manufactured with the addition of thermophilic lactic acid bacteria (*Lb. delbrueckii* ssp. *bulgaricus* and *Lb. helveticus*) compared to cheese with the addition of *Lactococcus* genus bacteria and NSLAB. The high proteolytic activity of thermophilic LAB compared to non-starter LAB and bacteria of the genus *Lactococcus* may result from their stronger peptidase activity and activity against casein by releasing its amino acids which with their share are transformed into other compounds that develop the sensory traits of cheeses. The analyzed cheese models manufactured with the addition of *Lb. helveticus* LH-B01 were characterized by a higher proteolytic activity compared to both the cheese models with the heat-treated adjunct XT–312 starter and control cheeses. Undoubtedly, it results from the fact that they were manufactured with the use of *Lb. helveticus* LH-B01 which is known as one of the most proteolytic strains among LAB [28].

Ardö et al. [37] reported that additional bacterial cultures of *Lb. helveticus* heat-treated at 67 °C for 10 s caused intensification of the proteolysis and flavor traits of low-fat Cheddar cheeses. Usually, the addition of various heat-treated cultures of lactic acid bacteria in cheese making was intensifying proteolysis and had a positive effect on the aroma traits, sometimes inducing bitterness in ripened cheeses [4].

Herreros et al. [38] demonstrated that the proteolytic activity of *Leuconostoc mesenteroides* subsp. *mesenteroides* bacteria was lower than that of bacteria of the genus *Lactococcus lactis* subsp. *lactis* but higher compared to the activity of most of the analyzed LAB. A similar activity of *Leuconostoc* genus bacteria isolated from Manchego cheese was reported by Nieto-Arribas et al. [39].

Data available from the literature show great differences in the proteolytic activity of microorganisms classified as LAB. These great differences are due to various properties of LAB, which affects many possibilities of their choice as adjunct cultures. These properties are strain-dependent, which may pose difficulties in the selection of appropriate composition of microorganisms. Various dynamics of proteolytic processes during cheese ripening results, most probably, from the vast diversity (combination) of basic and adjunct starters used in their manufacture, from diversity of NSLAB, as well as from initial pH and water content.

The application of adjunct bacterial cultures both in the form of fermentation-active cheese culture *Lb. helveticus* LH-B01 and long heat-treated fermentation-inactive XT–312 starter increased the proteolytic activity in the analyzed cheese models, which indicates the acceleration of the ripening process. Results achieved in the study enable concluding that acceleration of cheese ripening process occurs also in the case of relatively intensive heat treatment of adjunct starter (65–75 °C/15 min). Also compared to respective studies presented in literature, where adjunct starters were usually heat-treated at 50–70 °C but in a relatively short time (i.e., a few or ten to twenty seconds). Moreover often, inactivated lactobacilli are used as adjunct starter but not starter culture consisting of various strains of LAB as in the case of our study [4]. Relatively long heat treatment of the adjunct starter (ten to twenty minutes) has some practical advantage. Such a heat treatment may be easily and practically cost-free conducted in a tank used for sourdough preparation. The use of very short periods of heat treatment of adjunct starters requires equipment operating in the flow-through system, which is linked with investments, requires automation and may cause additional losses. Furthermore, the use of heat-treated adjunct starter could result in more rapid lysis of active cells of the basic starter, which also intensified proteolytic processes in the ripening process of model cheeses.

### 3.3. Determination of ACE Inhibitory Activity

ACE-inhibitory activity of experimental cheese models with additional heat-treated XT–312 culture and with adjunct active LH-B01 culture ripened for five weeks are presented in Figure 2.

Experimental cheese models containing active or inactive additional cultures had higher ACE-inhibitory activities compared with control cheese model at the end of ripening (five weeks). Following the proteolysis pattern as describe before, ACE inhibitory activities increased significantly (*p* < 0.05) with prolonged ripening. ACE inhibitory activity was the highest at the fifth week of ripening for the cheese model with LH-B01 culture and XT–312 starter heat treated at 65 °C. The increase in ACE inhibitory activities during ripening is in accordance with our previous findings [15,16] and others [31,40,41,42].

### 3.4. Microbiological Analysis

The total bacteria count (TBC) in the analyzed cheese models fitted within the range from 8.91 to 10.28 log CFU/g cheese on the first day after manufacture and from 8.90 to 9.91 log CFU/g after five weeks of ripening (Table 4).

Both the one-day cheeses and five-week-ripened cheeses manufactured with the adjunct culture *Lb. helveticus* were characterized by the highest TBC. In turn, the control cheeses showed higher TBC over the entire ripening period compared to the cheese with the heat-treated adjunct XT–312 starter. Stronger proteolysis in the cheese models with the heat-treated adjunct XT–312 starter compared to the control cheeses resulted, most likely, from increased mass of bacterial cells already after their manufacture and from more rapid apoptosis of bacterial cells, their lysis and release of proteolytic enzymes.

A similar dependency was determined in the case of the count of *Lactococcus* genus bacteria. The highest count of these bacteria was found in the cheeses with the addition of *Lb. helveticus* LH-B01 compared to the variants with the heat-treated adjunct XT–312 starter throughout the ripening period. The lowest count of *Lactococcus* spp. after five weeks of ripening was reported in the cheese models with the adjunct XT–312 starter heat-treated at 70 and 75 °C. In all variants of cheese models, the count of bacteria of the genus *Lactococcus* decreased insignificantly after five weeks of ripening compared to the count determined after one day of ripening. It may result from changes in water activity or the presence of phages responsible for a decrease in the population number of *Lactococcus* [1]. In the model cheeses with the heat inactivated adjunct starter, the count of *Lactococcus* genus bacteria was decreasing along with an increasing temperature of adjunct starter heating.

The initial count of non-starter lactic acid bacteria (NSLAB) in the analyzed cheeses ranged from 2.13 to 3.93 CFU/g cheese, whereas after five weeks of ripening it increased to 7.87–9.01 CFU/g cheese. The higher count of NSLAB at the end of ripening period was observed in the control cheese models and cheese models with the addition of *Lb. helveticus* LH-B01 compared to the heat-treated adjunct XT–312 starter.

NSLAB are lactic acid bacteria that constitute cheese-contaminating microflora and usually originate from raw milks and re-infection after milk pasteurization [43]. It was demonstrated that a low number of NSLAB may survive the pasteurization process, develop during ripening and reach the value of 6–8 log CFU/g cheese depending on ripening time and temperature [44,45]. During cheese ripening, the NSLAB develop more rapidly than the starter culture and become the predominating microflora after autolysis of starter cells [46]. The NSLAB include mainly heterofermentative mesophilic lactic acid bacilli (old species name: *Lactobacillus casei*—new name: *Lacticaseibacillus casei*, old name: *Lactobacillus paracasei*—new name: *Lacticaseibacillus paracasei*, old name: *Lactobacillus rhamnosus*—new name *Lacticaseibacillus rhamnosus,* and old name: *Lactobacillus plantarum*—new name: *Lactiplantibacillus plantarum*) although the presence of pediococci, *Leuconostoc* and micrococci is also reported. The initial count of NSLAB differs in various cheeses. In the case of Swiss type cheeses, the count of NSLAB reaches 10 CFU/g cheese and increases during ripening to 6 log CFU/g cheese after 10 weeks. The Cheddar cheeses contain even 8 log CFU NSLAB/g cheese immediately after ripening, whereas after 10 weeks of storage their count decreases by 2 logarithmic orders [43,44,47]. Starter applied in cheese making affects the growth rate and final count of NSLAB during ripening. Lactic acid bacteria were capable of growing on lysates of *Lactococcus* cells, which suggests that the lysis of starter cells may provide nutrients for NSLAB growth in cheeses. Some authors observe that a high content of free amino acids synthesized by peptidases released after starter cell death may stimulate the growth of NSLAB [48]. Sparse studies are available on a correlation between starter cultures and NSLAB that would confirm the hypothesis that lytic starter strains improve the development of NSLAB. Lane et al. [49] compared, among other things, the effect of starter bacteria strains on the growth rate of NSLAB in Cheddar cheeses. The NSLAB were developing faster in the cheeses manufactured with starter strains of *L. lactis* subsp. *lactis* ML3 and 303 compared to these produced with *L. lactis* subsp. *cremoris* AM1 and AM2. Considering results obtained in the present study, it may be observed that the heat-treated adjunct XT–312 starter affected a decrease in NSLAB count along with an increase in temperature of its heating. The cheese models with the addition of heat-treated starter were characterized by a lower count of NSLAB compared to both control cheeses and cheeses manufactured with the addition of a *Lb. helveticus* LH-B01, which may indicate that the adjunct starter was inhibiting the development of not only the basic active starter but also of NSLAB. The highest count of both NSLAB, TBC and *Lactococcus* spp. bacteria was reported in the cheese models with the addition of a cheese culture *Lb. helveticus* LH-B01, which may show that this culture stimulated the growth of starter and adjunct microflora in the analyzed cheese models and, perhaps, the development of more appropriate conditions for their growth.

## 4. Conclusions

The heat-treated adjunct bacterial cultures and the cheese culture *Lb. helveticus* did not affect the change in the composition of cheese model. The application of heat-treated adjunct XT–312 starter and *Lb. helveticus* LH-B01 was increased the proteolytic activity in the analyzed cheese models which indicates accelerating the cheese ripening. It was determined that the heat treatment of the adjunct starter even at higher temperatures (65 °C) and for 15 min effectively increased ACE inhibitory activity in the examined cheese models. In addition, the variants of cheese models with the heat-treated adjunct starter were characterized by lower TBC, NSLAB count and count of bacteria of the genus *Lactococcus* compared to the control cheeses, which may indicate that the heat-treated adjunct starter was inhibiting the growth of not only the basic active starter but also of NSLAB. It should be concluded that the heat-treated adjunct XT–312 culture and *Lb. helveticus* LH-B01 may be used as functional ingredients in Dutch-type cheese to maintain sufficient bioactive properties and improve proteolysis.

## Figures and Tables

**Figure 1 animals-11-02699-f001:**
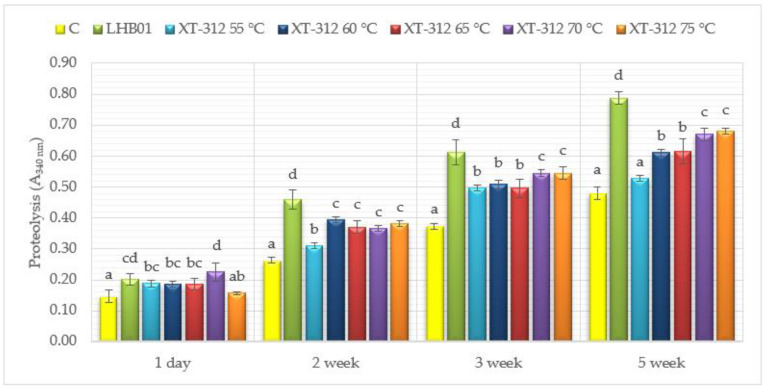
Proteolysis during ripening of Dutch-type cheese models (mean values and standard deviations, a–d homogenous groups, *p* < 0.05, *n* = 6).

**Figure 2 animals-11-02699-f002:**
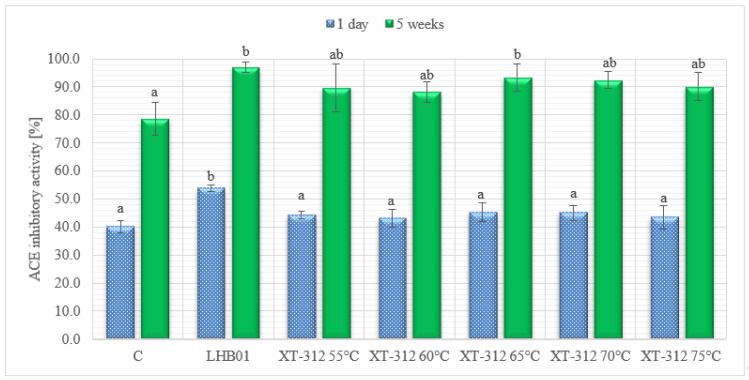
ACE inhibitory activity (%) of the experimental cheese models during ripening (mean values and standard deviations, a,b homogenous groups, *p* < 0.05, *n* = 6).

**Table 1 animals-11-02699-t001:** Cheese model variants.

Cheese Model Variant	Coagulating Enzyme	Basic Starter	Additional Starter
C	Fromase 2200 TL	2.0% CHN–19	(-)
LHB01			1.5% *Lactobacillus helveticus* (LHB01)
XT–312 55 °C			1.5% XT–312 starter heated at 55 °C/15 min
XT–312 60 °C			1.5% XT–312 starter heated at 60 °C/15 min
XT–312 65 °C			1.5% XT–312 starter heated at 65 °C/15 min
XT–312 70 °C			1.5% XT–312 starter heated at 70 °C/15 min
XT–312 75 °C			1.5% XT–312 starter heated at 75 °C/15 min

(-)—not added, C—control cheese model.

**Table 2 animals-11-02699-t002:** Selected physicochemical traits of cheese models after 5 weeks of ripening.

	Cheese Model						
	C	LHB01	XT–312 55 °C	XT–312 60 °C	XT–312 65 °C	XT–312 70 °C	XT–312 75 °C
Moisture (%)	46.31 ^a^	46.76 ^a^	47.02 ^a^	46.78 ^a^	46.89 ^a^	47.08 ^a^	46.90 ^a^
Moisture on fat free basis (MFFB %)	55.81 ^a^	56.46 ^a^	56.99 ^a^	56.66 ^a^	56.73 ^a^	57.00 ^a^	56.83 ^a^
Fat (%)	17.03 ^a^	17.18 ^a^	17.50 ^a^	17.44 ^a^	17.35 ^a^	17.40 ^a^	17.48 ^a^
Fat in dry matter (FDM %)	31.72 ^a^	32.26 ^a^	33.03 ^a^	32.76 ^a^	32.66 ^a^	32.87 ^a^	32.91 ^a^
pH	5.33 ^a^	5.31 ^a^	5.28 ^a^	5.32 ^a^	5.31 ^a^	5.29 ^a^	5.32 ^a^
Total N (TN %)	31.01 ^a^	31.04 ^a^	30.96 ^a^	30.31 ^a^	31.18 ^a^	31.00 ^a^	31.02 ^a^

^a^ Means with different superscript letters in line are statistically significant (*p* < 0.05, *n* = 6). C—control cheese model, LHB01—cheese models with adjunct *Lb. helveticus* LH-B01, XT–312 55–75 °C—cheese models with adjunct starter XT–312 heated at 55, 60, 65, 70 or 75 °C.

**Table 3 animals-11-02699-t003:** The change of ripening parameters of experimental cheese models.

Cheese Model Variant	Parameter (%)	Time of Ripening (Weeks)		
		1	3	5
Control	SN/TN	9.00 ± 0.14 ^a^	15.49 ± 0.15 ^a^	24.58 ± 0.11 ^a^
	TCA-SN/TN	6,69 ± 0,22 ^B^	8.74 ± 0.07 ^B^	16.06 ± 0.17 ^A^
	PTA-SN/TN	0.14 ± 0.02 *^A^*	0.29 ± 0.09 *^A^*	1.03 ± 0.11 *^A^*
LHB01	SN/TN	8.74 ± 0.12 ^a^	21.16 ± 0.60 ^c^	30.56 ± 0.44 ^b^
	TCA-SN/TN	7.53 ± 0.23 ^C^	15.92 ± 0.13 ^E^	19.19 ± 0.18 ^E^
	PTA-SN/TN	0.19 ± 0.03 *^A,b^*	1.73 ± 0.12 *^C^*	3.16 ± 0.16 *^B^*
XT–312 55 °C	SN/TN	8.69 ± 0.13 ^a^	16.02 ± 0.14 ^a^	25.53 ± 0.75 ^a^
	TCA-SN/TN	6.11 ± 0.19 ^A^	7.57 ± 0.09 ^A^	16.43 ± 0.16 ^A^
	PTA-SN/TN	0.11 ± 0.03 *^A^*	0.25 ± 0.07 *^A^*	1.13 ± 0.17 *^A^*
XT–312 60 °C	SN/TN	9.82 ± 0.18 ^b^	16.23 ± 0.27 ^a^	25.45 ± 0.71 ^a^
	TCA-SN/TN	6.84 ± 0.16 ^BC^	8.48 ± 0.13 ^B^	16.99 ± 0.21 ^B^
	PTA-SN/TN	0.16 ± 0.02 *^A^*	0.62 ± 0.10 *^B^*	1.27 ± 0.08 *^A^*
XT–312 65 °C	SN/TN	10.02 ± 0.21 ^b^	15.28 ± 0.17 ^a^	24.20 ± 0.91 ^a^
	TCA-SN/TN	7.09 ± 0.20 ^B,C,D^	11.90 ± 0.08 ^D^	18.30 ± 0.11 ^D^
	PTA-SN/TN	0.25 ± 0.02 *^B^*	0.80 ± 0.10 *^B^*	1.28 ± 0.08 *^A^*
XT–312 70 °C	SN/TN	10.18 ± 0.21 ^b^	17.18 ± 0.14 ^b^	24.82 ± 0.36 ^a^
	TCA-SN/TN	7.11 ± 0.20 ^B,C,D^	10.37 ± 0.16 ^C^	17.82 ± 0.13 ^C^
	PTA-SN/TN	0.13 ± 0.03 *^A^*	0.76 ± 0.06 *^B^*	1.34 ± 0.14 *^A^*
XT–312 75 °C	SN/TN	10.62 ± 0.19 ^b,c^	17.68 ± 0.50 ^b^	24.53 ± 0.50 ^a^
	TCA-SN/TN	7.27 ± 0.14 ^C,D^	10.25 ± 0.28 ^C^	17.66 ± 0.15 ^C^
	PTA-SN/TN	0.15 ± 0.04 *^A^*	0.81 ± 0.14 *^B^*	1.36 ± 0.09 *^A^*

^A,B,C,D,E,a,b,c,*A,B,C*,^ Values within the same parameter in column marked with the same superscript lowercase or uppercase letter are not significantly different at *p* < 0.05. SN/TN: soluble nitrogen per total nitrogen; TCA-SN/TN: nitrogen soluble in trichloracetic acid per total nitrogen; PTA-SN/TN: nitrogen soluble in 5% phosphotungstic acid per total nitrogen; TN: total nitrogen.

**Table 4 animals-11-02699-t004:** Count of selected groups of bacteria in the analyzed cheese models during ripening.

		Cheese Model						
		C	LHB01	XT–312 55 °C	XT–312 60 °C	XT–312 65 °C	XT–312 70 °C	XT–312 75 °C
Total bacteria count (TBC) (log CFU/g)	1 day	9.95 ^b^	10.28 ^b^	9.07 ^a^	8.91 ^a^	9.14 ^a^	8.99 ^a^	8.97 ^a^
	2 week	10.86 ^c^	10.22 ^b^	9.48 ^a^	9.39 ^a^	9.52 ^a^	9.53 ^a^	9.30 ^a^
	3 week	9.96 ^bc^	10.30 ^c^	9.19 ^a,b^	9.16 ^a^	9.23 ^ab^	9.25 ^ab^	9.38 ^a,b^
	5 week	9.42 ^b^	9.91 ^c^	8.92 ^a^	9.02 ^a^	8.92 ^a^	8.90 ^a^	8.96 ^a^
Count of *Lactococcus* spp. (log CFU/g)	1 day	9.31 ^d,e^	9.48 ^e^	9.03 ^c,d^	8.72 ^b,c^	8.51 ^b^	8.70 ^bc^	8.09 ^a^
	2 week	9.55 ^b^	9.46 ^b^	8.73 ^a^	8.43 ^a^	8.67 ^a^	8.59 ^a^	8.53 ^a^
	3 week	9.44 ^b^	9.63 ^c^	8.77 ^a^	8.88 ^a^	8.89 ^a^	8.86 ^a^	8.95 ^a^
	5 week	8.07 ^a,b^	8.91 ^b^	7.96 ^a,b^	8.22 ^a,b^	8.23 ^a,b^	7.87 ^a^	7.85 ^a^
Count of NSLAB (log CFU/g)	1 day	3.34 ^d^	3.93 ^e^	3.84 ^e^	2.77 ^c^	2.35 ^a,b^	2.13 ^a^	2.63 ^bc^
	2 week	6.04 ^a^	6.44 ^a^	5.35 ^a^	5.33 ^a^	5.56 ^a^	5.34 ^a^	5.49 ^a^
	3 week	8.60 ^b^	8.79 ^b^	7.64 ^a^	7.63 ^a^	7.74 ^a^	7.51 ^a^	7.70 ^a^
	5 week	9.01 ^b^	8.87 ^b^	8.17 ^a^	8.25 ^a^	7.96 ^a^	8.09 ^a^	7.87 ^a^
Count of *Lb. helveticus* LH-B01 (log CFU/g)	1 day	-	6.53	-	-	-	-	-
	2 week	-	8.02	-	-	-	-	-
	3 week	-	7.35	-	-	-	-	-
	5 week	-	6.76	-	-	-	-	-

^a,b,c,d,e^ Means with different superscript letters in line are statistically significant (*p* < 0.05, *n* = 6). C—control cheese model, LHB01—cheese models with adjunct *Lb. helveticus* LH-B01, XT–312 55–75 °C—cheese models with adjunct starter XT–312 heated at 55, 60, 65, 70 or 75 °C. (-) not determined.

## Data Availability

The data presented in this study are available on request from the corresponding author.

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
