# Peer review of "The Impact of the Adjunct Heat-Treated Starter Culture and Lb. helveticus LH-B01 on the Proteolysis and ACE Inhibitory Activity in Dutch-Type Cheese Model during Ripening"

_animals, 2021, doi:10.3390/ani11092699_

Round 1

Reviewer 1 Report

Comments to the Author

GENERAL COMMENTS

This study aimed to evaluate the effect of adjunct heat-treated starter XT-312 and a cheese culture Lb. helveticus LH-B01 on the proteolytic activity and angiotensin converting enzymes inhibitors (ACE) in cheese models during ripening.

The idea represents an interest in the cheese industry field especially hard cheese types.

My recommendation on submitted manuscript is to be accepted after minor revisions.

The comments and questions provided below may be considered as a guide to the authors to put their work into more appropriate form for publication.

SPECIFIC COMMENTS

- L. lactis is more common than Lc. lactis as an abbreviation for Lactococcus lactis.

Leu. mesenteroides instead of L. mesenteroides

Kindly revise throughout the MS

- In tables, letters indicating significance need to be put in superscript

- Manuscript needs language editing as many grammar and structure mistakes were noticed, e.g.

  • L410 correct to increased
  • L411 cheese ripening
  • L418 it is either “In summary” or “It can be concluded that the heat..”, both sentences together is not convenient.

- References need to be updated as only 3 references of 45 references are within the recent 5 years.

Author Response

Dear Reviewer,

Thank you very much for reviewing our manuscript: The impact of the adjunct heat-treated starter culture and Lb. helveticus LH-B01 on the proteolysis and bioactive compounds in Dutch-type cheese model during ripening. We have adopted all your suggestions.

Your suggestions have seriously contributed to the improvement of our manuscript. All changes compared to the original version have been highlighted in blue. Hope the revised manuscript will be evaluated as improved, in any case, we are willing to consider any further request.

SPECIFIC COMMENTS

L. lactis is more common than Lc. lactis as an abbreviation for Lactococcus lactis.

Leu. mesenteroides instead of L. mesenteroides

Kindly revise throughout the MS

Changed in modified text.

- In tables, letters indicating significance need to be put in superscript

Changed in modified text.

- Manuscript needs language editing as many grammar and structure mistakes were noticed, e.g.

  • L410 correct to increased

Changed in modified text.

  • L411 cheese ripening

Changed in modified text.

  • L418 it is either “In summary” or “It can be concluded that the heat..”, both sentences together is not convenient.

Changed in modified text.

- References need to be updated as only 3 references of 45 references are within the recent 5 years.

Changed in modified text.

Reviewer 2 Report

This manuscript is written in an easy-to-follow, pedagogical style. For each feature observed in the results, all possible explanations are clearly laid out. Presented results seem to be of good quality, and explained carefully. However, I have only a few concerns mentioned below.

1) In the title of the manuscript authors mention the bioactive compounds in Dutch-type cheese model but have not analyzed such as acids, aldehydes, esters, ketones etc.

2) Authors should include biochemistry profiles or gel electrophoresis profiles of heat-treated starter culture and Lb. helveticus LH-B01

3) Analysis of peptides after proteolysis

Author Response

Dear Reviewer,

Thank you very much for reviewing our manuscript: The impact of the adjunct heat-treated starter culture and Lb. helveticus LH-B01 on the proteolysis and bioactive compounds in Dutch-type cheese model during ripening.

Your suggestions have seriously contributed to the improvement of our manuscript. Hope the revised manuscript will be evaluated as improved, in any case, we are willing to consider any further request.

  • In the title of the manuscript authors mention the bioactive compounds in Dutch-type cheese model but have not analyzed such as acids, aldehydes, esters, ketones etc.

We changed title on: The impact of the adjunct heat-treated starter culture and Lb. helveticus LH-B01 on the proteolysis and ACE inhibitory activity in Dutch-type cheese model during ripening.

  • Authors should include biochemistry profiles or gel electrophoresis profiles of heat-treated starter culture and Lb. helveticus LH-B01

Yes we agree that it could have been done, in order to show a relationship between the biochemical profile in the heat-treated starters and the profile of the obtained cheese models. However, not necessarily such a relationship in the obtained cheeses will be present. Therefore, testing the suggested hypothesis seems interesting and needed and will be included in our future research.

  • Analysis of peptides after proteolysis

We would like to kindly thank you for your valuable suggestion. We admit that mentioned researches would enriched our manuscript and we will include them in our currently researche. However all results related to submitted manusript are finished, all materials have been utilized and we don’t have any possibility to supply this research with new tests. 

Reviewer 3 Report

The paper “The impact of the adjunct heat-treated starter culture and Lb. helveticus LH-B01 on the proteolysis and bioactive compounds in Dutch-type cheese model during ripening” contributes to the growth of literature for dairy specialists,  as well as food producers offering functional products, especially dairy products (milk drinks or cheeses).

Before  the manuscript acceptation for publication in “Animals” the following items should be revised:

Materials and Methods

The authors did not write the number of repetitions of an experiment (series of samples).

Results

Table 3

“Values within the same parameter in the column marked with the same letter” -  Uppercase or Lowercase?

Conclusions

The authors did not write whether there is a need for sensory analysis.

Author Response

Dear Reviewer,

Thank you very much for reviewing our manuscript: The impact of the adjunct heat-treated starter culture and Lb. helveticus LH-B01 on the proteolysis and bioactive compounds in Dutch-type cheese model during ripening. We have adopted all your suggestions.

Your suggestions have seriously contributed to the improvement of our manuscript. All changes compared to the original version have been highlighted in yellow. Hope the revised manuscript will be evaluated as improved, in any case, we are willing to consider any further request.

Materials and Methods

The authors did not write the number of repetitions of an experiment (series of samples).

Changed in modified text (lines 110-111).

Results

Table 3

“Values within the same parameter in the column marked with the same letter” -  Uppercase or Lowercase?

Changed in modified text (lines 226-227).

Conclusions

The authors did not write whether there is a need for sensory analysis.

Yes we agree, there is a need for sensory analysis, but no longer on cheese models only on cheese. We are continuing such research.
